

**PeerJ Hubs**
Published on behalf of

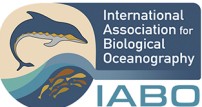

International Association for Biological Oceanography
IABO

# Pulsed sounds caused by internal oxygen transport during photosynthesis in the seagrass *Halophila ovalis*

Hin-Kiu Mok[1,2,3], Yen-Wei Chang[4], Michael L. Fine[5], Keryea Soong[1], Yu-Yun Chen[6], Richard G. Gilmore[7], Linus Yung-Sheng Chiu[8], Shi-Lin Hsu[9] and Hai-Jin Chang[1]

[1] Department of Oceanography, National Sun Yat-Sen University, Kaohsiung, Taiwan
[2] University Malaysia Sarawak, Institute of Biodiversity and Environmental Conservation, Sarawak, Malaysia
[3] National Museum of Marine Biology and Aquarium, Pingtung, Taiwan
[4] Seafood Technology Division, Fisheries Research Institute, Ministry of Agriculture, Keelung, Taiwan
[5] Department of Biology, Virginia Commonwealth University, Richmond, VA, United States
[6] Department of Sport Management, Aletheia University, New Taipei City, Taiwan
[7] Estuarine, Coastal and Ocean Science, Inc., Vero Beach, Florida, United States
[8] Institute of Undersea Technology, National Sun Yat-Sen University, Kaohsiung, Taiwan
[9] Marine Ecology and Conservative Research Center, National Academy of Marine Research, Kaohsiung, Taiwan

Corresponding author
Yen-Wei Chang,
ywchang@mail.tfrin.gov.tw

## ABSTRACT

Oxygen bubbles that leak from seagrass blades during photosynthesis have been hypothesized to cause cavitation sounds in aquatic plants. Here we investigate low-amplitude sounds with regular pulse rates produced during photosynthesis in seagrass beds of *Halophila ovalis* (Qitou Bay, Penghu islands and Cigu Lagoon, Taiwan). Sound pulses appear in the morning when illumination exceeds 10,000 Lux, peak at midday and decrease in midafternoon on a sunny day. Frequencies peak between 1 to 4 kHz, durations range between ca. 1.8 to 4.8 ms, and sound pressure level 1 cm from the bed is 105.4 ± 0.5 dB re 1 μPa (1100 h on a cloudy day). Sounds attenuate rapidly with distance, disappearing beyond 15 cm. Blocking sunlight or administering herbicide stops ongoing sounds. Gas bubbles are not typically seen during sound production ruling out cavitation, and external force (scissor cutting or plant pressed against the substrate) applied to the patch, leaves, petioles, or rhizomes generally increases pulse rate. We suggest sound emission is caused by internal oxygen transport through pores in diaphragms (a whistle mechanism) at the leaf base and nodes of the rhizome.

# INTRODUCTION

Seagrasses have aerenchyma tissues with continuous air-filled lacunae in leaves, rhizomes, and roots (*Armstrong, 1979*; *Larkum et al., 1989*; *Borum et al., 2006*; *McKenzie, 2008*). The lacunae allow movement of photosynthetic oxygen produced below the leaf epidermis to flow to petioles, stem, rhizomes, and roots (*Roberts, McComb & Kuo, 1984*; *Lee et al., 2023*)

by phase diffusion (*Sorrell & Dromgoole, 1987*). The lacunae in leaf and rhizome are connected and have diaphragms at the nodes and transitional regions (*Larkum et al., 1989*). The diaphragms are perforated, containing 0.5–1.0 µm interstitial pores (*Roberts, McComb & Kuo, 1984*) that allow gas transport. Most seagrasses, unlike terrestrial angiosperms, lack stomata (*Roberts & Caperon, 1986*), thereby preserving oxygen although some is lost by diffusion through the thin cuticle.

During high levels of photosynthesis, increased oxygen pressure in the lacunae can lead to a doubling of the leaf volume, and a continuous stream of bubbles will escape from cuts on the seagrass surface (*Roberts & Caperon, 1986*). Likewise, turtle grass *Thalassia testudinum* leaves swell, doubling above their early morning volume, due to production of photosynthetic gases (*Zieman, 1974*). He commented that "by early afternoon on a shallow calm *Thalassia* bed with little water flow, the bed can readily be heard to hiss from the rapid bubbling" and "gave the appearance of a "newly open: bottle of beer." Further, *Felisberto et al. (2015b)* noted ambient noise from a *Posidonia oceanica* seagrass bed exhibited a diurnal pattern with most energy between 2–7 kHz.

*Borum et al. (2006)* determined that the largest loss of oxygen from seagrasses is from leaves to the water column during periods of high light. The continuous leakage of oxygen from roots and rhizomes to the anoxic sediment both during light and dark periods also represents a major oxygen sink. As substantial production of $O_2$ occurs during photosynthesis, *Felisberto et al. (2015a)* suggested that the bursting stream of bubbles emerging from the seagrass leaf may be responsible for producing sounds and that a passive acoustic system could be used to monitor the $O_2$ production of a seagrass meadow. During photosynthesis in submerged waterweed, *Elodea canadensis*, gas bubbles are generated, and a series of sound pulses can be produced when the bubbles emerge from the stomata or damaged tissue; however, these sounds are not caused by the passage of the bubbles in the water (*Kratochvil & Pollirer, 2017*). The Canadian pondweed *Elodea canadensis*, generates popping sounds when gas bubbles leak from stomata, producing sounds with frequencies mainly below 10 kHz, although energy might extend to the ultrasonic range (*Kratochvil & Pollirer, 2017*). In seagrass, mass flow of oxygen could theoretically occur on a small scale driven by internal pressurization from photosynthesis or by leaf movement due to external physical force (*Borum et al., 2006*). However, no one has reported specific sounds generated by such movement although (*Kratochvil & Pollirer, 2017*) pointed out that at high temperatures additional high, harmonic sounds with strong continuous frequency changes occur. They suggested these sounds might be related to the passage of fine bubbles through vessels of different cross sections.

During an acoustic survey of spawning aggregations of sciaenid fishes in the Indian River Lagoon, Florida, U.S.A. continuous series of click sounds on sunny days in meadows of Manatee Grass, *Syringodium filiforme* and Shoal Grass, *Halodule wrightii* were noticed (HK Mok, 1980, personal observations). We find similar sounds in meadows of other seagrasses on the Southwestern coast of Taiwan, Penghu Island and Dongsha Atoll in the South China Sea; sounds of these species were short pulses with varying frequencies and pulse rates (HK Mok, 2008, personal observations).

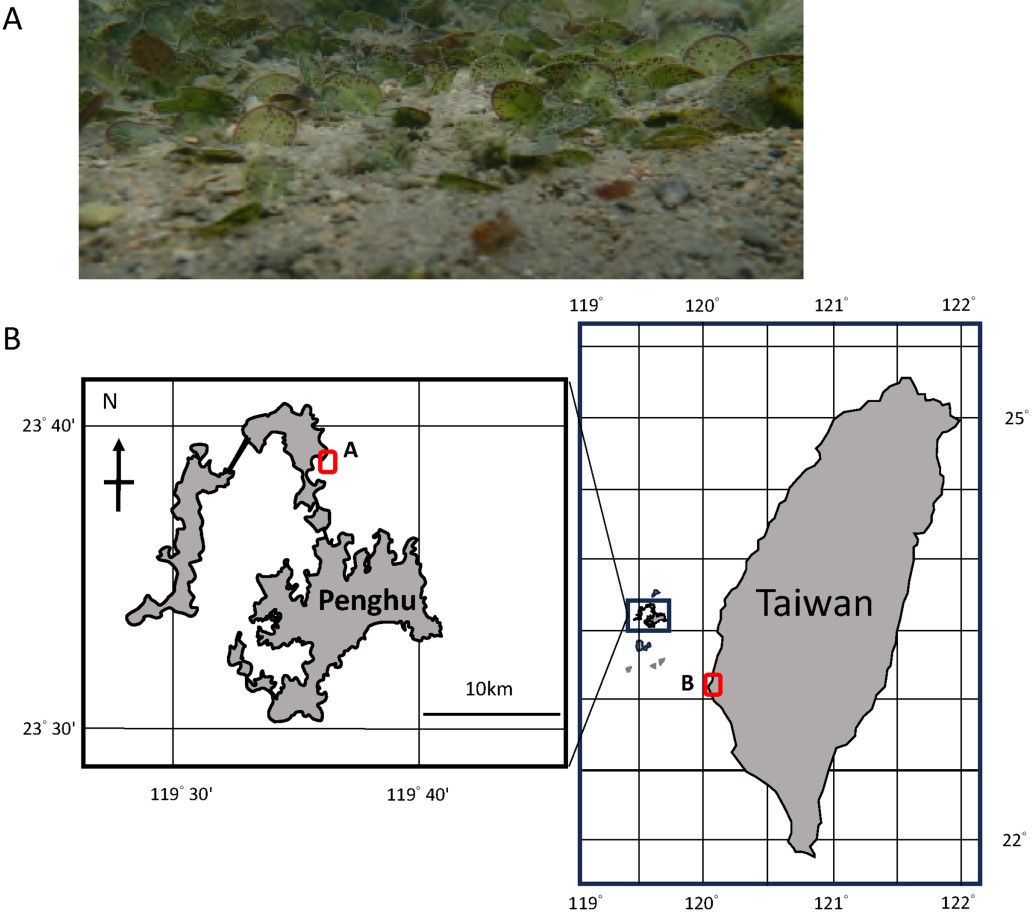

**Figure 1 Locality of the study site.** *In-situ* view of *Halophila ovalis* patch in Qitou Bay (A), and locality of the study sites in Qitou Bay, Penghu islands and Cigu Lagoon (B). This map was adapted from Map data © 2024 Google (https://www.google.com.tw/maps/).

In this study we describe the sounds produced by *Halophila ovalis* (Fig. 1A), their diurnal periodicity, test the effect of light intensity, block light and expose seagrass to a photosynthetic blocker inhibiting sound production. We hypothesize that pulsed sounds are caused by flow of oxygen gas in the lacunae, which passes through the pores on the diaphragms and test this hypothesis experimentally.

## MATERIALS AND METHODS

Portions of this text were previously published as part of a preprint (https://doi.org/10.1101/2024.05.14.594143).

Dugong (spoon) grass, *Halophila ovalis* (Hydrocharitaceae) was studied because of its small size compared with other seagrasses in Taiwanese waters and its accessibility for experimentation (Fig. 1A). It occurs in the Indo-Pacific region (*Kuo, 2020*) and is abundant in Qitou Bay (23°38′55″N 119°36′19″E) Penghu Islands (Fig. 1B).

Diel ambient sound was monitored on 24–25, July, 2020 and 25, August, 2021. Depth at the recording site at high tide was ca. 2 m and about 10 cm at low

tide. Additionally, experiments were conducted at Qitou Bay and at Cigu Lagoon (23°09′18.2″N 120°06′18.4″E) on the southwest coast of Taiwan during low tide when the seagrass patches were submerged at depths <10 cm (also see below).

Pulse series and individual pulses are sorted based on pulse rate and power spectral density level. Hourly changes in pulse-series types, pulse rate and pulse number in daytime were estimated by sampling hourly from recordings between 0600 to 1800 h on 07/25/2020. Pulse rate was determined from four 5-s sections from the first 5 min in each quarter hour, *e.g.*, 16 sections per hour. Means were and standard deviation were used to estimate pulses per hour. Pulse duration was measured for the first six cycles in the pulse because subsequent cycles tend to decrease in amplitude and were hard to recognize.

Sounds were recorded from six patches of *H. ovalis* and six control sites in nearby sandy areas without vegetation at Qitou Bay, and amplitude was measured at Cigu Lagoon during low tide in sunny conditions and at night. The sound pressure level was estimated from a patch of *H. ovalis* with a calibrated system placed on the substrate within 1 cm of the leaves.

At the *H. ovalis* meadow at Qitou Bay from 07/24–25/2020, illumination intensity ranged from 0–103,137 lux, and temperature ranged from 27.2–42.5 °C. Maximum temperature occurred at 1126 h, and the period of high temperature (above 40 °C) lasted from 0941–1129 h. High tides occurred at 1308, 0806 and 1357 h, and low tides at 1934, 0806, and 2020 h.

To test if sound emission is related to photosynthesis, three patches of seagrass were collected, and substrate sediment and invertebrates were gently removed. Samples with wet weights of 121, 110 and 132 g respectively (estimated number of leaves 2,330, 2,356 and 2,889) were placed in separate plastic trays (45 cm × 35 cm × 12 cm) filled with seawater to a depth of ca 10 cm. The trays were placed on the shore in close proximity, and an omnidirectional hydrophone was placed sideways on each patch. Light intensity and temperature were recorded in one tray. Sound was monitored continuously during four 15 min treatments: control (normal sunlight condition); trays covered by a transparent plastic plate, by a black plastic plate (dark condition) and then uncovered, a return to normal sunlight. Finally, we added a photosynthesis inhibitor (Diuron dissolved in absolute alcohol to the seawater at 0.01 g per liter) and recorded sounds for 30 min.

To evaluate the relationship between sound pulse emission rate and light intensity, a patch of *H. ovalis* was placed in a 29 cm × 17 cm × 24 cm glass aquarium set in the laboratory with 10 L of artificial seawater. A plant grow-light set (Smart Grow Light, SJZX-ZH001; with eight brightness levels) faced the front of the aquarium 30 cm from the HOBO behind the tank. Sounds were recorded for 5 min at five brightness levels, 10%, 30%, 50%, 70% and 100%; the recorded illumination intensity at 100% was 16,870 Lux. The hydrophone was suspended in the middle of the aquarium.

To test if sound emission and acoustic characteristics are related to oxygen transport from the leaves to the rhizomes and roots, the following experiments were conducted.

(1) Trays were monitored by video camera, hydrophone, and light intensity gauge in the morning as sunlight increased. To monitor potential bubble release during photosynthesis activity at dawn (*e.g.*, 0630 to 0800 h), a video camera was placed adjacent to the

hydrophone and the seagrass patch. Similar video and audio monitoring was also conducted on undisturbed plants in the field.

(2) Sound was monitored in an intact patch of seagrass in the meadow in which leaf was compressed by fingers and the patch was pressed against the substrate.

(3) A hydrophone was placed on a patch of seagrass in Qitou Bay to record sounds for 5 min after which individual leaves were cut slowly and transversely from the petiole below its junction with the leaf (diaphragm therefore intact). The detached leaves were left in the vicinity of the petiole, where any sounds would be recorded. This procedure was repeated three times.

## Recording systems

We used two Sony PCM-M10 digital recorders (frequency response 20 Hz to 20 kHz), and one Sony PCM-A10 High-Resolution Audio Recorder (frequency responses linear PCM 44.1 kHz, 24 Bit, from 40 Hz–20 kHz). Hydrophones (Aquarian Audio H2A, frequency range: <10 Hz to >100 kHz). A third recorder was placed inside a water-tight housing. Finally, a calibrated recording system including an Aquarian Audio H2A hydrophone connected to a Sony D100 PCM Recorder (linear PCM 192 kHz, 24 Bits) was used to measure sound pressure level. For continuous recording of illuminance and temperature, a HOBO pendant MX temperature/light data logger (MX2202; temperature: 20 °C to 70 °C; light: 0-320,000 lux) was used.

Diel changes in sounds from an *H. ovalis* patch at Qitou Bay were tracked with the water-tight housing recording system and a HOBO for temperature and light data for 24 h.

## Pulsed sound analysis

Avisoft-SAS Lab Pro 5.2.08 and MATLAB were used for signal analysis. The recorded wave files (sampling rate at 44.1 kHz) were down-sampled at 12 kHz. Acoustical parameters including the pulse rate (number of pulses per second, in Hz, for those pulse series with a homogenous pulse rate), number of pulses per 1,200 s, inter-pulse interval, power spectral density, spectral centroid (which measures the spectral position and shapes and considered as the center of 'gravity' of the spectrum) and peak amplitude (mV) were measured from oscillograms, spectrograms and power spectra. Sound pressure levels were measured using Raven pro 1.5.2. Temporal spectral changes in underwater soundscapes were analyzed using a software available in Google Colab (https://colab.research.google.com/drive/1Kd94iI2WjhT8KeGSQx4FeDdaD7o_gjlU). Sound pressure level of the pulsed sounds was measured using Raven Pro 1.6. (known quantity pressure = 162.1 dB; 0 to peak) and MATLAB with window length = 0.0058 s (256 points), overlap = 0.0051 s (226 points).

## Statistical analysis

Comparisons between sound parameters of the seagrass sounds produced under different environmental conditions were made using Mann-Whitney U Test by the analytical software XLSTAT in Excel. Significant level was set at $p < 0.05$.

## RESULTS

### Pulsed sound characteristics

Pulsed series with regular pulse rates occur primarily during the day and decrease in midafternoon and then more markedly in late afternoon and evening (Figs. 2, 3A). In the evening and early morning (0500 and 0600 h), pulses occur irregularly.

Series of pulses with stable pulse rates occur between 0700 to 1700 h. They were sorted into high-rate and low rate series with pulse rates respectively ca. >38 pulses/s and <18 pulses/s. Low pulse rates can be further sorted into three sub-categories by power spectral density, namely, L1, most energy below 2 kHz; L2, energy peaks at ca. 2 kHz; L3, energy from ca. 0.5 to 5 kHz. Acoustic energy in the high-rate pulses is concentrated at ca. 2 and 3 kHz (Fig. 2). Pulse durations (measured for the first six cycles) of these four pulse types are 2.3 ± 0.3 ms (1.8–2.9), $n$ = 30 (H); 3.8 ± 0.5 ms (3.2–4.8), $n$ = 38 (L1); 3.3 ± 0.2 ms (3.1–3.8), $n$ = 31 (L2); 3.0 ± 0.4 ms (2.4–3.9) $n$ = 84 (L3) (Fig. 2).

Temporal occurrence of series types varies. The high-pulse rate series occurs between 0900 to 1200 h, with peak at 1200 h (Figs. 2, 3). In the low-rate series, sub-categories L1 and L2 appear between 0700 and 1400 h, whereas subcategories L3 appears from 0700 to 1600 h (Fig. 2). Series L1, L2, and L3 peak at 1000, 1100, and 0700 h, respectively (Fig. 2). Times between 1100 and 1300 h included more low-rate pulses than other daytime hours (Fig. 2).

On sunny days, a patch in Qitou Bay produced series of high-amplitude broad-band pulsed sounds (500–6,000 Hz) with stable inter-pulse intervals (*e.g.*, 6 pps) mixed with weaker but high-repetition-rate pulses (as much as 89 pps) with peak frequencies at 2,000 and 3,000–4,200 Hz. Duration of high-amplitude pulses averaged 2.3 ± 0.3 ms (range: 1.8–2.9 ms; $N$ = 30) compared to 3.3 ± 0.5 ms (range: 2.4–4.8 ms; $N$ = 153) (Mann-Whitney U Test; $p < 0.00001$) for the weaker high rate series. In some instances, only series with pulses at different repetition rates occurred. During periods with bright light, pulse rates ranged from 16 to 89 pps; and a high pulse rate series could last for 200 s before tapering and then stopping. Daytime pulses, increase from 1 to ca. 90 pulses per sec with light intensity and often have a regular pulse rate. Maximal pulse rate occurred between 1200 and 1300 h.

Pulse series with a single peak at 1.4 kHz diminished at ca. 1530 h when the light intensity was 18,304 lux. Between 1630 and 1820 h energy decreased at the dominant frequency peak (ca. 1,930 Hz)—the main energy peak for photosynthesis pulses (see below).

Pulses with two different rates sometimes appear simultaneously during the day at high light levels (Fig. 2). Series can start with a high pulse rate and then decrease or vice versa without changes in light intensity (see below), suggesting variable rates of photosynthetic gas production or a lag between light intensity and oxygen production.

Irregularly-paced pulses also occur at night, *e.g.*, 0100 to 0200 h ca. 611.9 ± 123.2 pulses per 5 min; range ca. 443–804 pulses; $N$ = 7). Those occurring between 0100 and 0400 h contributed high acoustic energy compared to other parts of the evening and early morning and had a higher spectral centroid located between ca. 2,700 and 2,800 Hz

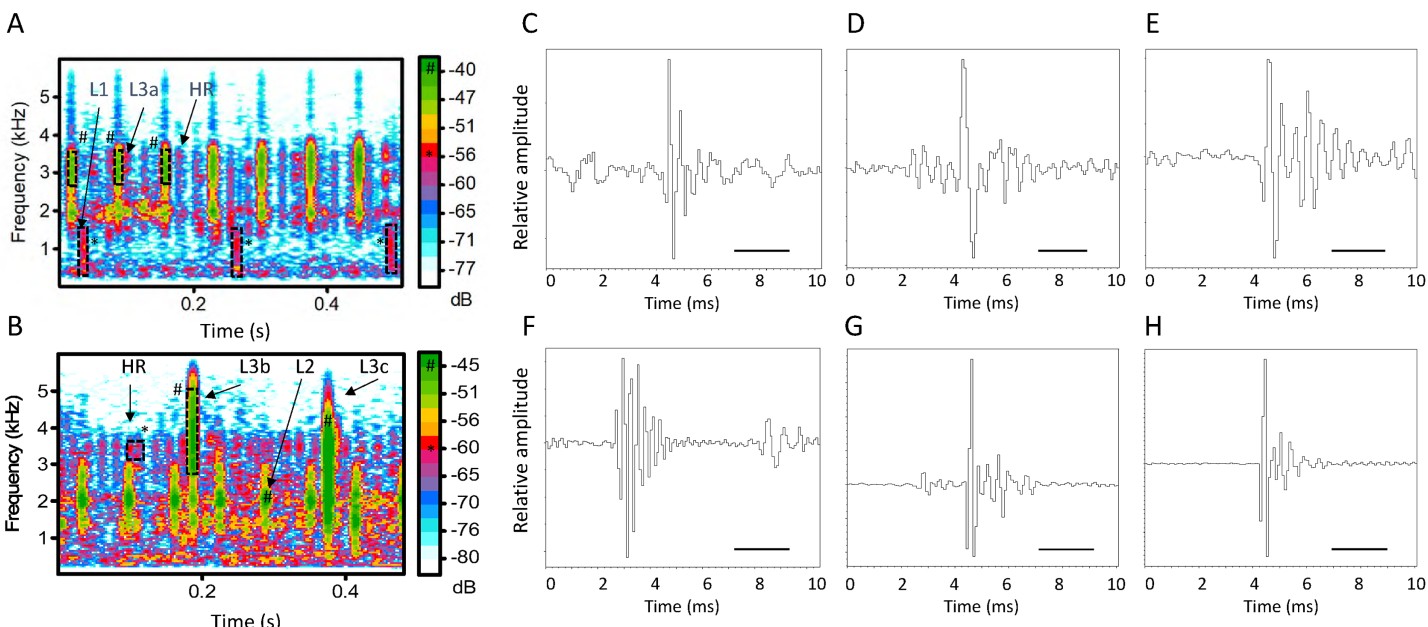

**Figure 2** Two spectrograms of the pulsed sound recorded at 1000 h (A) and 1200 h (B) from the *Halophia ovalis* patch in Qitou Bay showing arrangement of pulse series with different pulse rates and spectral densities. Arrows point to pulse types of which the expanded waveforms are shown in (C–H). FFT = 256, frame length 100, overlap 88%. HR: High-rate pulse; Low-rate pulse includes L1: Acoustic energy mainly <than 2 Khz; L2: Acoustic energy mainly distribute around 2 kHz; L3a: energy broadly spread between ca. 500 Hz to 4.5 kHz, with energy peaks at ca. 2.0 and 3.0 kHz; L3b: with low energy bands in its frequency range; L3c; without obvious peaks in the frequency range. Scale bar = 2.0 ms. A number sign (#) represents the green color (below 100 dB); an asterisk (*) represents the red color (approximately 140 dB).

(Table 1), and a bigger energy difference occurred (*i.e.*, 2.1 to 2.9 dB) between the dominant and secondary frequency peaks (ca. 1,930 and ca. 2,700 Hz, respectively). The pulses at this period are relative high pitched (Table 1).

Waveforms are variable; the first six pulses in all the pulse were relative easy to recognize. The second cycle is often but not always weaker than the following cycles (Figs. 2D, 2E). The amplitude rises abruptly with a peak at the first, second, or third cycle, followed by variable decay (Figs. 2C–2H). However, in most of the pulses analyzed from a sunny day, the first cycle has the highest amplitude and the second cycle is often weaker than the adjacent cycles (Figs. 2G, 2H). Pulses with energy more homogeneously spreading across the frequency range show a smooth decay in amplitudes (Fig. 2F). In night time, early morning and late afternoon pulses, the second or third cycle reaches the maximum amplitude of the pulse, whereas the first cycle in the pulses under bright light has the maximum amplitude (Figs. 2G, 2H).

The mean sound pressure levels (SPL) of a high-amplitude pulse series with a pulse rate of ca. 14 pps was 105.4 ± 0.5 dB re 1 µPa (range: 105.1–105.8; $N$ = 30), (hydrophone 1 cm from the seagrass patch, light intensity 50,000 to 60,000 Lux on a sunny day and low tide). The SPLs were obtained by calculating the mean squared sound pressure, with a window length of 0.0058 s (256 data points) and an overlap of 0.0051 s (226 points) given a sampling frequency of 44.1 kHz. The peaks of the sound pressure level series were identified, and the mean values of these peaks were averaged. The sound amplitude

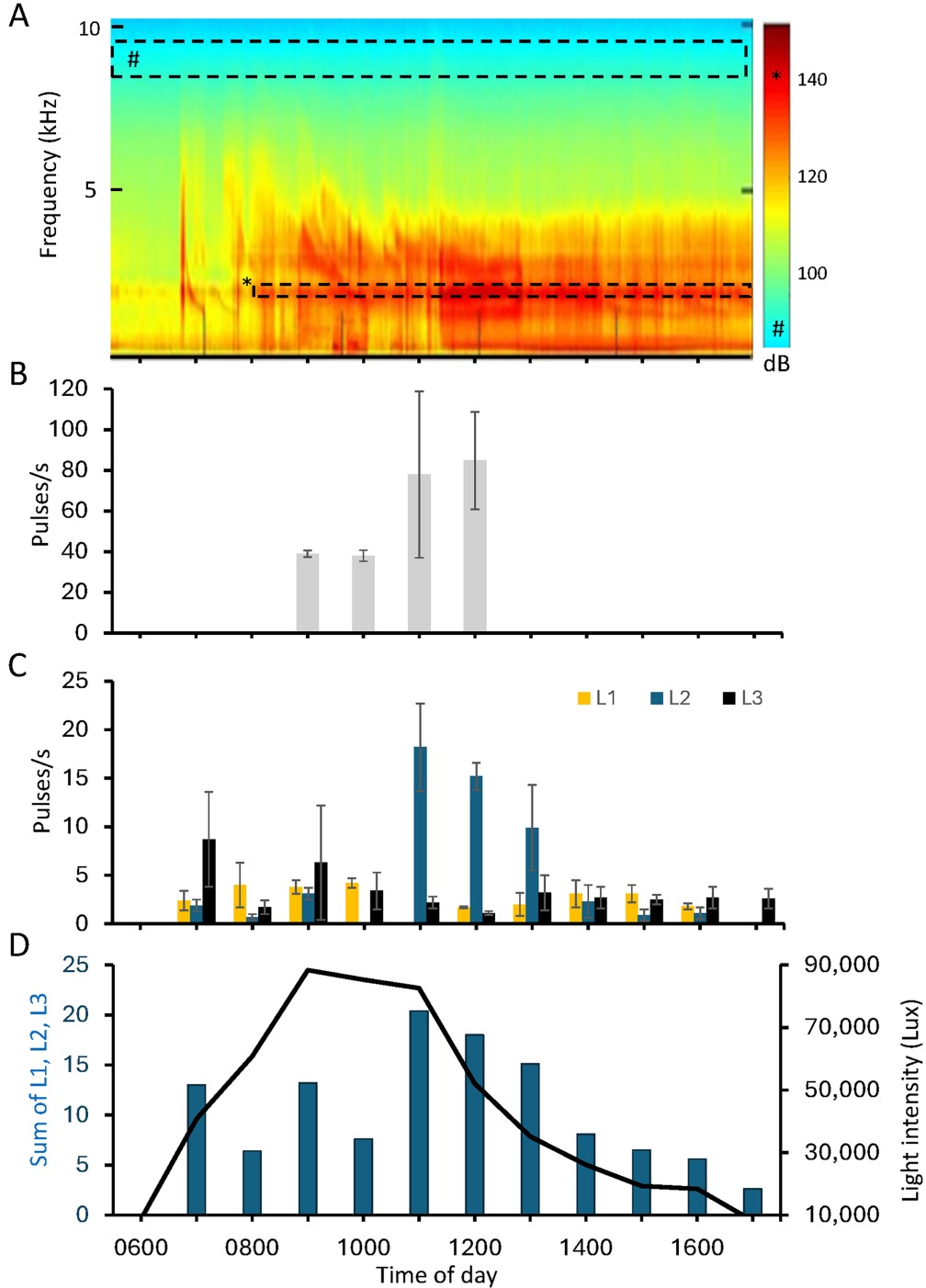

**Figure 3** **Temporal analysis of ambient noise and pulse types at *Halophia ovalis*.** (A) A median-based spectrogram showing the temporal power spectral on daytime of 25/07/2020 for the ambient noise at *Halophia ovalis* patch in Qitou Bay with maximum sunlight intensity at around 1100 h, low and high tides at ca. 0830 and 1400 h. Hourly occurrence of the pulse types in the daytime shown in bar charts. (B), Type H; (C), L1, 2, 3; (D), Average pulse rates and average sunlight intensity (line). H: high-pulse rate series; L1: low-pulse rate series, with most energy below 2 kHz; L2: low-pulse rate series, with energy spread adjacent to 2 kHz; L3: low- pulse rate series, energy spread from 0.5 to ca. 5 kHz, with variable power spectral density, or peaks at ca. 2 and 3 kHz. A number sign (#) represents the green color (below 100 dB); an asterisk (*) represents the red color (approximately 140 dB).

**Table 1 Spectral centroid and difference between the amplitudes for the dominant-frequency peak at ca. 1,960 Hz and secondary frequency peak at ca. 2,900 Hz.**

| Time of day (h)[*] | Spectral centroid (Hz) | Amplitude (dB) at ca. 1,960 Hz higher than that at 2,700 Hz |
|---|---|---|
| 1820–1920 | 2,830 | 7.1 |
| 1920–2020 | 2,850 | 6.0 |
| 2020–2120 | 2,900 | 6.8 |
| 2120–2220 | 2,620 | 7.7 |
| 2220–2320 | 2,560 | 8.1 |
| 2320–0020 | 2,530 | 7.4 |
| 0020–0120 | 2,670 | 6.0 |
| 0120–0220 | 2,820 | 2.5 |
| 0220–0320 | 2,800 | 2.9 |
| 0320–0420 | 2,810 | 2.1 |
| 0420–0520 | 2,740 | 2.6 |
| 0520-0620 | 2,630 | 6.1 |
| 0620–0720 | 2,870 | 5.4 |
| 0720–0840 | 2,660 | 8.4 |
| 0840–0940 | 2,600 | 6.9 |
| 0940–1040 | 2,510 | 2.1 |
| 1040–1140 | 2,430 | 5.6 |
| 1140–1240 | 2,440 | 5.1 |
| 1240–1340 | 2,210 | 8.1 |
| 1340–1440 | 2,270 | 8.3 |
| 1440–1540 | 2,270 | 8.2 |
| 1540–1640 | 2,240 | 6.4 |
| 1640–1740 | 2,470 | 4.7 |
| 1740–1840 | 530 | 3.9 |

Note:
[*] Measured from the long-term (1-h) average spectra from 1820/24/07/2020 to 1820/25/07/2020. Frequencies become lower in high light levels.

attenuated rapidly over open sandy areas and no periodical pulses indicative of photosynthesis sounds were recorded at 20 cm from three patches; these sound might have decayed into the background noise (Fig. 4A). The pulsed sounds can be barely heard when an observer's ear is next to the seagrass patch (HK Mok, 2017, personal observation).

## Experiments on pulsed sound production due to photosynthesis
### Sound and light intensity

In samples in plastic trays under bright sun, pulsed sounds with stable inter-pulse interval were present, indicating functional photosynthesis on carefully removed plants. These sounds rule out the possibility of production by marine invertebrates cryptically associated with the rhizomes and roots.

Pulse rates in bright direct-light (50,000 and 68,000 lux) did not change when trays were covered by a transparent plate (401 ± 72.7 and 402 ± 46 pulses/5 min, $N = 3$). Pulses

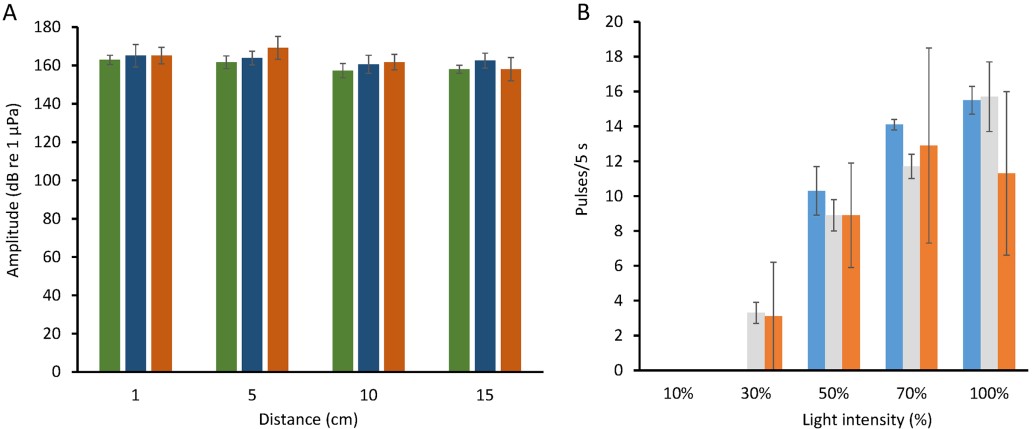

**Figure 4 Analysis of amplitude attenuation and pulse frequency in relation to brightness levels in *Halophila ovalis*.** (A) Attenuation of amplitude (mean and SD) of pulsed sounds from the edge of *Halophila ovalis* patches. Three replicates, each with 30 measurements. (B) A bar chart showing relation between number of pulses/5 s (mean and standard deviation) at five brightness levels in three trials tested in an aquarium set on the bench in the laboratory. Illuminance at 10%, 30%, 50%, 70%, and 100% were 790, 3,970, 7,440, 10,270 and 16,870 Lux, respectively. Notice absence of pulsed sound at the 10% brightness level.

immediately vanished after the tray was covered with a black plastic plate, during which luminance inside the tray ranged from 0 to 0.43 Lux and reappeared ca. 90 s after plate removal. Pulses then regained their normal repetition rate in ca. 173 s. Therefore, light is required for pulse production, and pulse rate increased with light intensity (Fig. 4B).

In sound emission tests on an intact seagrass patch at Cigu Lagoon (60,000 and 70,000 Lux in light and 80 Lux in dark), pulse rate decreased from $298.7 \pm 43.7$ pulses/30 s ($N = 6$) to 0 pulses in less than 1 min when covered with a black plastic sheet. A replicate on a near-by patch likewise decreased similarly from $277.5 \pm 39.8$ pulses/30 s ($N = 6$) to 0 pulses/30 s.

Pulse rate decreased 30 s after Diuron addition and vanished completely after about 23 min in three trials. Light intensity at the bottom of the plastic container varied between ca. 50,000 to 60,000 Lux.

### Pulsed sounds and bubbles leaking from the seagrass

On one occasion under sunlight conditions, we saw gas bubbles emerging from a small patch of *H. ovalis* placed in a beaker. Pulse rate was $11.8 \pm 3.4$ pulses/5 s or 2.4 pulses/s ($N = 14$). Pulse duration was $11.1 \pm 1.4$ ms (range 10.1–15.9 ms), with three pronounced frequencies at ca. 1.0, 2.5, 5 kHz. Maximum amplitude of a pulse occurred in the third cycle. Therefore, gas bubbles leaking from the seagrass can generate sounds, but they were different in duration, spectral density and waveform from typical photosynthesis pulses (*i.e.*, longer duration, more cycles per unit time, and amplitude peaked at the third cycles instead of the first cycle). Streams of minute air bubbles were noted during brightness testing in an aquarium, but no sounds with similar frequency profile of the photosynthesis pulses were recorded adjacent to the bubble streams. These findings indicate that typical sounds produced by these plants are not related to bubble production.

***Sound emission and acoustic characteristics are related to oxygen transport in the air-channel system of H. ovalis.***

In cloudy conditions, when pulses had an irregular emission rate, force applied by fingers and the seabed typically (but not always) increased pulse repetition rate and peak frequency followed by a decrease. Sounds were composed of high and low amplitude pulses with variable repetition rates. High amplitude pulses had repetition rates of 44.9 ± 22.2 pulses/s (range 18–87 pulses/s; $N = 15$) *vs.* 155.7 ± 45.6 pulses/s (range 89–230 pulses/s; $N = 7$) for low amplitude ones. Sound emission sometimes changed from a mixed series to one with only low amplitude pulses suggesting that high-amplitude pulses require greater leaf pressure. Duration of pulse series averaged 19.9 ± 20.2 ms (range 6.8 to 32.9 s; $N = 11$).

Cutting a petiole below its junction to the leaf also increased repetition rate likely because the force applied generated gas movement inside the lacunae. Pulses from a seagrass patch decreased as the leaves were cut below the leaf-petiole junction or from the middle of the leaf one by one. It took about 120 s after which no pulses were noticed even though leaves remained in the vicinity of the seagrass patch. Repeated tests conducted in the trays and in the field obtained similar results. No visible gas bubbles were noticed leaking out from the cut.

## DISCUSSION

Gas bubbles with sounds emitted from *H. ovalis* in a beaker were seen once and were associated with pulsed sounds (HK Mok, 2020, personal observation); but pulsed sounds were not recorded from stream of small bubbles from *H. ovalis* in a small tank. However, pulses were recorded repeatedly from undisturbed *H. ovalis* patches in shallow water, during which no bubbles were seen. Therefore, bubbles are not the source of these sounds, supporting *Kratochvil & Pollirer*'s *(2017)* suggestion of sounds caused by internal gas transfer. *Kratochvil & Pollirer (2017)* reported gas bubble sounds with most energy below 10 kHz and a dominant frequency around 4 kHz, from submerged Canadian waterweed, *Elodea canadensis* in a small tank. These sounds ceased after lights were turned off. They also pointed out that at high temperatures, besides sounds exiting from the stomata, other high, harmonically structured sounds with strong, continuous frequency changes occur, and these sounds might be related to the passage of fine bubbles through vessels of different cross sections.

During light periods, $O_2$ pressure increases within *H. ovalis* leaves and causes oxygen flow to lower parts of the plant that inhabit an oxygen poor substrate (*Borum et al., 2006*). Pulsed sounds appear in series with irregular paces when morning illumination increases above 10,000 Lux. As light intensity increases and leaves swell from gas accumulation, pulse rate increases and becomes regular. Under bright sunlight, low amplitude pulse series emerge and mix with the high-amplitude series. Under this situation, an external force applied to leaves boosts only the rate of the low-amplitude series but not the loud series, suggesting that separate threshold gas pressures are necessary to produce low and high-amplitude pulses. Photosynthetic pulses begin to decrease around 1500 h although light intensity is still high. Fewer pulses likely reflect indicate decreased photosynthesis, which might be due to photoinhibition, temperature stress, or high carbohydrate

end-product inhibition of photosynthesis (*Larkum, Drew & Ralph, 2006*; *Azcón-Bieto, 1983*; *Goldschmidt & Huber, 1992*). Therefore, we suggest that pulsed sounds from plant beds could be used to reveal the health of seagrass meadows and environmental stresses. *Kratochvil & Pollirer (2017)* pointed out that analyses on the acoustic effects during photosynthesis of aquatic plants open completely new research opportunities for plant physiology. As photosynthesis might be interfered by thermal, water turbidity, pollutant stresses and sound production is related to photosynthesis, sounds could also be a useful indicator for environmental stress to the seagrass through interfering with photosynthesis.

Plants covered and held in dark conditions stop producing sounds, but similarly produce high-pulse rate and low-amplitude sound series several seconds after return to high luminance. These continue and are followed by the addition of the slow rate high amplitude pulses. Pressing the patch at night or in a tray covered with black plastic does not generate pulsed sound.

Cutting the seagrass petioles or pressing the patch between fingers and seabed in sunlight rapidly increases pressure in the lacunae, and based on *Fletcher (1992)* we suggest plants produce a whistle-type sound by causing gas to escape through the 0.5 and 1 μm holes in the diaphragms between the leaf and stem. Fletcher notes that "the basis of aerodynamic sound arises from a steady flow that becomes unstable if it is forced to change direction or to undergo shear deformation near a solid surface. In this case gas pressure from photosynthetic oxygen in relatively wide lacunae will change direction as it passes through 0.5 to 1 μm pores in the diaphragms (*Roberts, McComb & Kuo, 1984*). The system is further complicated by multiple pores per diaphragm and multiple lacunae per leaf. Therefore, pressurized gas is typically required for sound emission (but see below), and we suggest higher gas pressure thresholds for loud than quiet pulses. Warmer temperatures should also cause a minor increase leaf pressure and hence sound repetition rate and amplitude since it will increase gas volume within the leaves (Charles' Law) and therefore flow through the diaphragms.

Under high illumination, a series of pulses has a stable inter-click interval, implying pulses come from individual sources in variable acoustic states rather than multiple leaves or plants simultaneously. Termination of a series of intense pulses from a leaf would stop when pressure decreases below threshold. Subsequent intense sounds would then likely come from another leaf.

*Felisberto et al. (2015b)* recorded ambient noise at night with most energy between 2 to 7 kHz from a *Posidonia oceanica* seagrass bed at 2 to 20 m depths. The mean noise power was negatively correlated with dissolved $O_2$, and they suggest that power is correlated with photosynthetic activity, which is unlikely since photosynthesis would not peak at night. Our night recordings (0100 to 0300 h) also indicate sounds in the typical frequency band at ca. 2–3 kHz, which is presently unexplained.

Seagrass meadows attenuate sound rapidly compared to open shallow habitats (*Chang et al., 2019*; *Wilson, Wilson & Dunton, 2012*). The gas within adjacent seagrass leaves dissipates the acoustic-energy, and spreading of the sounds from a small bundle of seagrass should be minimal because of absorption by adjacent seagrass bundles.

The evolution of plants sounds is intriguing and unexplored. The ability to produce sounds in *Halophila ovalis* shares characteristics with evolutionary exaptations, a term coined by *Gould & Vrba (1982)* in which a functional character is later co-opted for a new use that enhances fitness. In acoustic communication the term has been invoked for the evolution of sound-producing structures in fishes (*Parmentier, Diogo & Fine, 2017*). For instance a tendon that causes rapid jaw slams in damselfishes (a feeding adaptation) has been turned into a sonic structure that functions in behavior (*Parmentier et al., 2007*), swimbladders that contribute to buoyancy and gas regulation have become sonic organs by attaching muscles that cause them to vibrate (*Fine & Parmentier, 2022*), and a defensive catfish spine that can be locked to deter predators (*Sismour et al., 2013*) has been modified into a stridulatory organ in which ridges on the spine base are rubbed against a rough surface on the pectoral girdle, a stick-slip mechanism as in a bow on a violin (*Mohajer, Ghahramani & Fine, 2015*). Similarly, oxygen produced in the leaves of *Halophila ovalis* causes them to swell, and the increased pressure drives gas transport through small holes in their diaphragms, pushing oxygen to underground portions of the plant in an anoxic environment. It is unknown whether these sounds provide signals to adjacent plants, which would be necessary to establish an exaptation. For now, therefore, we consider the sounds to represent an epiphenomenon related to gas transport rather than a functional acoustic signal.

## CONCLUSIONS

Active photosynthesis in *Halophila ovalis* increases leaf pressure (*Roberts & Caperon, 1986*), driving oxygen through pores in diaphragms that separate leaves from underground portions of the plant, which inhabit an anaerobic environment. Pressurized oxygen transport produces variable low rate and periodical sound pulses in the morning, and these become regular and more rapid as light intensity increases. Adding mechanical pressure to the leaves increases pulse rate, and dark exposure or a photosynthesis inhibitor stops sound production. We suggest the plant produces aerodynamic sounds when leaf pressure forces gas through submicron pores in the diaphragms separating parts of the plant.

## ACKNOWLEDGEMENTS

We are grateful to Prof. T. M. Lee, National Sun Yat-sen University who gave us advice in using herbicide Diuron and logistic support for field recording. Our sincere thanks go to Prof. Adair, B. for calling our attention to the presence of the paracrytic-type of stromata in *Thalassia testudinum*, its possible role in generating sound, and the value of acoustic monitoring for pollution impacts on the seagrass meadow. Dr. C. W. Chang, National Academy of Marine Research provided the calibrated recording system to measure sound amplitude. Mrs. P. H. Chiu, National Sun Yat-sen University, assisted us in measuring the sound pressure level. Prof. K. S. Chiu, National Kaohsiung University of Science and Technology, helped with field collection. Dr. T. H. Lin, Academia Sinica, kindly provided the software he developed that uses an algorithm for source separation models to visualize temporal spectral changes in underwater soundscapes (https://colab.research.google.com/drive/1Kd94iI2WjhT8KeGSQx4FeDdaD7o_gjlU).

### Funding

This study was supported by grants from The Ministry of Science and Technology (MOST), R.O.C. to Keryea Soong. There was no additional external funding received for this study. The funders had no role in study design, data collection and analysis, decision to publish, or preparation of the manuscript.

### Grant Disclosures

The following grant information was disclosed by the authors:
The Ministry of Science and Technology (MOST), R.O.C.

### Competing Interests

Richard G. Gilmore, Jr. is employed by Estuarine, Coastal and Ocean Science, Inc., Florida, U.S.A.

### Author Contributions

- Hin-Kiu Mok conceived and designed the experiments, performed the experiments, analyzed the data, prepared figures and/or tables, authored or reviewed drafts of the article, and approved the final draft.
- Yen-Wei Chang performed the experiments, analyzed the data, prepared figures and/or tables, authored or reviewed drafts of the article, and approved the final draft.
- Michael L. Fine conceived and designed the experiments, authored or reviewed drafts of the article, and approved the final draft.
- Keryea Soong conceived and designed the experiments, authored or reviewed drafts of the article, and approved the final draft.
- Yu-Yun Chen performed the experiments, authored or reviewed drafts of the article, and approved the final draft.
- Richard G. Gilmore conceived and designed the experiments, authored or reviewed drafts of the article, and approved the final draft.
- Linus Yung-Sheng Chiu conceived and designed the experiments, authored or reviewed drafts of the article, and approved the final draft.
- Shi-Lin Hsu performed the experiments, analyzed the data, prepared figures and/or tables, and approved the final draft.
- Hai-Jin Chang performed the experiments, analyzed the data, prepared figures and/or tables, and approved the final draft.

### Data Availability

The raw data is available in the Supplemental Files.

### Supplemental Information

Supplemental information for this article can be found online at http://dx.doi.org/10.7717/peerj.18114#supplemental-information.

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
