# Peer review of "Pulsed sounds caused by internal oxygen transport during photosynthesis in the seagrass Halophila ovalis"

_PeerJ, doi:10.7717/peerj.18114_

## Round 0.1 · original submission · Major Revisions

According to the reviewers, many aspects of your manuscript should be improved. First of all, and I think the most important, the use of citations and quoting in your paper. Please, be sure of what you report based on other papers. I don't want to be repetitive with what the reviewers wrote, so I have not so many comments apart the following:

Line 80: Please, enlarge the words into the maps (enlarge also the close up image of the Penghu Islands) and make them clearer and readable.
Line 85: The Qitou Bay and Cigu Lagoon are not reported in the map. Please, add them.
Line 101: "...three patches..." which was the magnitude of these patches? How many rhizomes per patch? How many leaves?
Line 110: "...a patch..." please, see my previous comment.
Lines 153-154: Why did use two online tools when you have some useful and more adopted and reliable free software available (e.g., R, PAST, Jamovi, etc). Even because I don't believe these online resources have tools to assess violation of heteroscedasticity, normal distribution or autocorrelation among samples. Can you repeat the analysis with more reliable software?
Lines 254-258: as said by one of the reviewer, why starting this way the discussion paragraph? Please, report first your own results and then what was found previously.

Reviewer 1 ·

Basic reporting

The authors describe the use of acoustic monitoring to experimentally determine the sound production mechanism by a seagrass (halophila ovalis) and its acoustic diel patterns. The experimental design appears to be good with the use of controls alongside experimental trays. The figures are good but I think that 9 is too many for the main text, perhaps the most important 3 -4 would be good. The statistical methods need more of an explanation in the methods section, e.g., why the selected methods were used etc. The introduction also needs a lot more background to provide context for the research. The conclusion would also benefit from more on the wider applications of the research and its value to the scientific community. Lots of style points throughout regarding consistency and clarity of language. Most sections would benefit from more subheadings.

Experimental design

The experimental design appears to be good with the use of controls alongside experimental trays. A sample size of six seagrass patches for each treatment seems sufficient. The figures are good but I think that 9 is too many for the main text, perhaps the most important 3 -4 would be good. The statistical methods need more of an explanation in the methods section, e.g., why the selected methods were used etc.

Validity of the findings

(see above and attached doc)

Annotated reviews are not available for download in order to protect the identity of reviewers who chose to remain anonymous.

Reviewer 2 ·

Basic reporting

This paper contains a lot of interesting data, from a number of different experiments, and a number of different sample preparations. It is on a topic that is of interest to the community and has the potential to be a useful contribution to the literature. There are a large number of ambiguities that the authors must address in order to fully convey what they have done. See the specific comments in the review section 4, additional comments.

Experimental design

The authors have reported a number of observations, and conducted some experiments, but this reviewer does not believe the conclusions of the paper have been proved. This is more specifically address in the comments provided in review section 4.

Validity of the findings

This reviewer does not criticize the basic data that the authors present, but does criticize the conclusions the authors draw, as described in the specific comments in review section 4.

Additional comments

General comment: The authors describe a number of the methods in one section and then report on a large number of results in a results section. There are also a number of different in situ sites and ex situ preparations. This format works well when there are only one or two different samples, or one or two different experiments, but here, it is difficult for the reader to know exactly which experimental materials and method was used for which result. In what follows, many specific ambiguities are described, an in addition to clearing up those ambiguities, I also think this paper would be far more useful to the reader to break it into sections that completely describe and report results for experiments conducted at one site, or in one of the several ex situ preparations (including trays, aquaria and beakers).

Lines 55-57: Authors state that Borum et al (2006) "...suggested that the bursting stream of bubbles emerging from the seagrass leaf may be responsible for producing sounds and that a passive acoustic system could be used to monitor the O2 production of a seagrass meadow." This reviewer did not find such an idea presented in Borum at al (2006). There is no mention of sound, noise, or acoustic signals in Borum et al (2006). Authors must be thinking of another paper, but I am not sure which one. Please revise.

Lines 61-62: Authors state: "The pondweed Potamogeton spp., generates ultrasonic popping sounds with energy between 18-25kHz via leaking gas bubbles from the stomata (Kratochvil and Polliver, 2017; Greenhalgh et al., 2023)."

Pondweed Potamogeton is not mentioned in Kratochvil and Polliver, 2017.

Note that although Kratochvil and Polliver (2017) state:

"The main energy (the place with the highest-level frequency components) of one part of the sound pulses lies in the ultrasonic range. (Figure 1)"

The data in Figure 1, which is for Elodea canadensis, is only shown on a frequency axis that terminates at 11 kHz, and the amplitude is diminishing below 9 kHz, which is not in the ultrasonic range. Another discrepancy in Figure 1 is that Kratochvil and Polliver state: "In most cases the average interval time between pulses is in the range of 1–2 msec (Fig. 1)." yet their Fig. 1 shows 25 events in one second, and hence the time between pulses is clearly more like 40 msec. I am not sure we can rely on their characterization of this data as being in the ultrasonic range. The data they show in their paper does not agree with their verbal assessment. Hence, I do not agree with using Kratochvil and Polliver (2017) as a citation for ultrasonic emission due to gas bubbles from aquatic plants. On the other hand, the web publication Greenhalgh et al. (2023) shows a figure with data that includes ultrasonic acoustic emissions from the pondweed Potamogeton. Suggest to find a citation for a peer-reviewed publication for either pondweed emissions or otherwise ultrasonic emissions from aquatic plants. Please revise these phrases that refer to ultrasonic acoustic emissions accordingly.


Lines 63-64: Authors state: "At high but unspecified temperatures, harmonic sounds with continuous frequency changes often occur (Kratochvil and Polliver, 2017). Kratochvil and Polliver suggested they are caused by internal gas flows."

Kratochvil and Polliver (2017) includes one paragraph about this: "At high temperatures, other, unusual sounds occur. These apparently reflect the passage of fine gas bubbles through vessels of different cross section. They are not related to gas bubbles exiting from the stomata because they also occur if no bubble series are visible. These are usually high, harmoniously structured sounds with strong, continuous frequency changes. The physiological causes of this noise remain unknown. Possibly these are cavitation noises caused by intern gas flows." They do not show any data to support these statements.

I find some discrepancy between Lines 63-64 and the actual statement from Kratochvil and Polliver (2017). First the word "often" is not used in the original. They state the cause is unknown but offer a possible source as internal gas flow. I suggest the authors summarize the Kratochvil and Polliver (2017) paragraph more accurately.


Lines 68-70: Authors state: "During an acoustic survey of spawning aggregations of sciaenid fishes in the Indian River Lagoon, Florida, U.S.A. (Mok and Gilmore, 1983), we recorded continuous series of click sounds on sunny days in meadows of Manatee Grass, Syringodium filiforme and Shoal Grass, Halodule wrightii"

There is no mention of the recordings of aquatic plants in Mok and Gilmore (1983) although the author's statement makes it sound like there are such recordings in that publication. Please revise the statement to indicated that sounds were recorded during the work that was discussed in Mok and Gilmore (1983) but those sounds were not presented or discussed in that publication.

Line 102: Authors report mass of samples. Useful to report more information about the samples, such as: How many plants? What was the area of the seabed that was harvested for the samples? About how many leaves per plant? What is the mass ratio of leaves to rhizomes? Are there any epiphytes?

Line 104: Authors state: "...and a hydrophone was placed on each patch." Please provide a more-complete description of how the hydrophone was placed in each patch. Was the hydrophone sitting on or touching plant matter? Was the hydrophone sitting on the surface of the tray? What was the orientation of the hydrophone relative to the tray? A figure would be helpful.

Figure 1: The text that is used to label the maps in Figure 1(B) is difficult to read and the resolution of the figure should be improved, or the text enlarged. It seems that one site is shown in Figure 1(B) but two sites are mentioned in the text around lines 81 through 85. Why not show the second site on the map?

Figure 1: Cigu Lagoon is one of the sites where data was collected, but it is not shown on the map, or if there is a label on the map, it is not easily visible. Please clearly indicate the locations where the data was collected.

Figure 2: The authors should clearly state the site that generated the data in Figure 2. Two (or maybe three) sites are mentioned in the text around lines 81 through 85 although that language is not clear. First the authors state in line 81 that spoon grass "...is abundant in Qitou Bay (23°38’55" N 119°36’19" E) Penghu Islands (Fig. 1B, C)." Then the authors state in line 84 and 85: "Additionally, experiments were conducted at Qitou Bay and at Cigu Lagoon (23°09’18.2"N 120°06’18.4"E)." It is not clear if there are one or two sites at Qitou Bay, since it is mentioned twice. Please clarify.

Figure 2: The tidal height plot looks very linear, in that every height change is the same as its neighbors, making a triangle wave plot. It seems tidal data is usually more sinusoidal. Can the authors comment on this in the text? Is this normal for the area?

Figure 2: Can the authors place subfigure (A) and subfigure (B) on time axes that are aligned and that have the same tic marks and tic labels? This way the reader can most-accurately relate the data in (A) to the data in (B).

Figure 2: Please clarify the amplitude of the the spectrogram in Figure 2(A). The color bar is labeled "dB." What is the reference? The abstract states that amplitudes on the order of 83 dB re 1 µPa were found, but the levels in this plot exceed 140 dB. If the absolute amplitude of the spectrogram is not known or used, then able with relative dB.

Figure 2: In other spectrogram plots, the authors provide details about the FFT length, frame length, overlap, etc. Please provide for the spectrogram in Figure 2.

Figure 3: Please state which site is responsible for this data.

Figure 3: There seems to be a discrepancy between the temporal metrics in the text and the axis labels in the plots of Figure 3. For example, pulse rates range from 38 pulses per second to 18 pulses per second (line 162), but in Figure 3(A), which has a time axis in units of milliseconds, we see about 7 pulses in about 0.5 milliseconds, which would be a pulse rate of about 14,000 pulses per second.

Figure 3: Please provide the dB reference for the colorbar.

Figure 4: Quoting from the caption: "A, H; B, L1; C, L2; D, L3a; E, L3b; F: L3c; G:" It is very difficult for the reader to sort out which pulse type is shown in this figure since one has to map the figure sub label (A, B, C, etc) through to the pulse type (H, L1, L2, etc). It would be far easier to directly label the pulse type in each sub-figure. You can keep the A, B, C, subfigure labels too, but adding the pulse type would short circuit the need for the reader to apply the mapping.

Figure 4: There is no amplitude information in any of the oscillogram plots. The vertical axes are only marked with the zero value. It would be very useful to provide at least one more mV marking on the vertical axes, so that one can get a sense of the amplitude differences between the types.

Figure 4: The dB reference should be stated. One suggestion would be to normalize all the spectrograms in this figure so that the maximum value across the entire set of spectrograms is set to zero dB. Then the relative amplitudes of all the pulses would be readily visible. Then a single color bar could be used and the seven others could be eliminated. This would make a lot of room so that the power spectrum plots could be made a little larger, which would then allow the font size on the power spectrum axes to be make large enough to see.

Lines 189-191 and Figure 6: Authors state: "Those occurring between 01:00 and 04:00 contributed high acoustic energy compared to other parts of the evening and early morning (Fig.6; Table 1)..." There is no data from 01:10 in Fig. 6. The data in Fig. 6 starts at 04:00. It seems the time interval (01:00 and 04:00) in the text is wrong.

Lines 194 and Figure 6: Authors state: "The pulses at this period are relative high pitched (Fig. 6; Table 1)." With this sentence, the text devoted to Fig. 6 ends, and except for the caption, there is no other description of the data presented in Fig. 6. Please include a more complete description of the data in Fig. 6 and its importance. Here are some questions: What is the meaning of the color bar, which is labeled "percentage." It is not at all clear what this means or how it was calculated. If it is a percentage of time when a particular spectrum occurs, then it seems unlikely that for every case, the percentage increases monotonically from low level to high level and entirely without frequency dependence. What are the units of amplitude on the vertical axes of each plot in subfigures A through D? There is poor overlap between the times in the four subfigures A through D and the data shown in subfigures E, F and G. For example, 04:00 and 06:00 are not included in the time series shown in E, F and G, which start at 07:00. What is the meaning of the colors (blue, red and gray) used in subfigure F?

Lines 203-207: Authors state they calculate sound pressure levels (SPLs). By definition, sound pressure level is a metric based on a root-mean-square time average of acoustic pressure. For a transient signal like the pulses seen here, the sound pressure level of a single pulse, or multiple pulses if included in the average, depends on the length of time that was used to calculate the time average. Therefore, the authors should report exactly how the average was taken. On the other hand, pulses are often characterized by using decibels calculated with peak pressures. If the authors are referring to dB based on peak pressures, then they are not formally SPLs, they are simply dB peak re 1microPa. Sometimes biological pulses (echolocation clicks, for example) are characterizes by peak-to-peak pressure. Please modify accordingly.

Line 208-209 and Figure 7: Authors report: "No signal was recorded at 20 cm from 3 patches (Fig. 7). The pulsed sounds can be barely heard when one’s ear is next to the seagrass patch (pers. obs.)." Then Fig 7 presents data from a propagation experiment. There is not enough detail presented about this propagation experiment for the sentence above and the data in Figure 7 to be very useful to the reader. Here are some questions: What is the geometry of the area where the propagation experiment was conducted? What is the depth? Is the boundary between the patch and the bare area a straight line or something else? A figure that explains this would be very useful. What is the material of the seabed in bare area adjacent to the patch? Do the authors expect sounds to be coming from a point in the patch, or from multiple points in the patch? The units shown in Figure 7 indicate that spectral density levels were used. What frequency band was used for this? The levels appear to drop about 5 dB in 15 cm, and then according to the text, no signal was recorded 5 cm further. The signal level at 15 cm is shown to be around 100 dB at 15 cm. It does not seem possible in open water, in absence of a barrier, or other absorbing object, for signals at 100 dB to become un-detectable another 5 cm away, unless they have fallen into the noise. If they are undetectable because they have fallen into the noise, then the authors should state that explicitly.

Line 213 and 220 and Fig.8 caption: Authors refer to "plastic trays" and "an aquarium" as the location where the data presented in Fig. 8 was acquired. According to the text in the paper, these are different experiments, hence there is some ambiguity about the location where this particular set of data was taken. Please clarify.

Figure 8: Why use the mapping of percentages to intensity? That is an extra layer between your reader and understanding the data. There is enough room in the horizontal axis to label the bars with the intensity in lux rather than in percentage, which is arbitrarily assigned to the highest used in the experiment. Higher intensities were reported for natural conditions, so this percentage label is somewhat misleading.

General comment about Results section B: Authors report the use of a opaque black plastic plate and refer to that as a "dark" condition, yet the authors have a light intensity instrument that was used to report the light intensity in the uncovered case. A finite-sized plate will not eliminate all the light on the patch in the tray. Can the authors report the intensity under the opaque plate? Although perhaps less important, the same could be said about the "transparent" plate, which is likely not truly "transparent" but instead most-likely attenuates the light intensity somewhat. The authors report the light intensity in the "dark" case one paragraph down, for the measurements done in situ. Why not for the measurements done in the trays (or aquarium)?


Line 229: Authors report about a "small patch" "placed in a beaker." The geometry of a patch and that of a beaker do not match. It seems more likely that some collection of plant tissue was placed in a beaker. Was this sample cleaned of sediment and infauna? Was natural water placed in the beaker too? Please describe this sample more completely.

Line 233-234. Authors report differences in sounds measured in beakers compared to sounds from "from typical photosynthesis pulses." It is not at all clear what is meant by typical. Are the authors comparing the sounds in a beaker from sounds in the trays or in situ? Please clarify. The sound recorded in any confined space, whether that be a tray or a beaker or a shallow sea is due to the source being convolved with acoustic response of the enclosed space. Hence a comparison between sounds recorded in different confined spaces may be more about the acoustics of the space than about the source itself. An analogy would be like saying human singing is different when it is produced in a large, hard-surfaced enclosure, such as a cathedral, than compared to the kind of singing produced in a small enclosure with soft surfaces, or outside on flat ground with no nearby boundaries. The singing is nominally the same, but the recorded sound is greatly affected by the environment. The authors at least need to acknowledge this when commenting on differences between sounds recorded in different enclosures, tanks, trays or in situ. One really does not know what is causing the difference. Is it the source, is it the environment, or some combination.

Line 235-236: Authors state: "Streams of minute air bubbles were noted during brightness testing in an aquarium, but no sounds were recorded adjacent to the bubble streams." Since bubbles that are emitted into water radiate sound with a frequency that is inversely proportional to the bubble size, it is quite likely that the limited bandwidth of the authors recording system is the explanation for why no sound was recorded. It would be easier to assess this if the authors could estimate the bubble size of these "minute" bubbles. The authors need to resolve this for this comment to be useful. In other words, this section can not be used as evidence that emission of minute streams is sound free.

Results reported in line 240 and below: Where was this data taken? Is it from a tray, from an aquarium, or in situ? Please clarify.


Line 240: Authors state: "...force applied by fingers to a patch..." It is not clear what this means. Are the authors squeezing plant tissue, are they putting pressure on plant tissue and compressing it between the fingers and the seabed? Please provide an unambiguous description. Is this at an in situ site or in one of the ex situ preparations?

Line 247: Authors state "cutting a petiole." Earlier, the authors described removing a leaf by cutting it off at the petiole. The distinction is important, because elsewhere in this paper, the authors relate how bubbles can be emitted from small openings, or "cuts," in plant tissue. Here, the authors refer to removing a leaf in the methods section. I presume that the data reported on line 247 is the result of removing a leaf by cutting it off at the petiole. Please clarify and eliminate this ambiguity for readers of line 247. This is further exacerbated by the text in the caption of the associated figure, Fig. 9, which states "...changes in pulse types...prior to...and during cutting..." which does not sound like a before and after leaf removal. These all need to be unified and unambiguously described.

Paragraph starting at Line 247: Please be more clear about what happens to the gas inside the plant tissue when the leaf is removed by cutting it off at the petiole. Were gas bubbles seen leaving the cut? If so, from what side, the leaf side or the stem side, or both? Were no gas bubbles seen?

Line 248: Authors state: "...Pulses from a seagrass patch decreased as the leaves were cut below..." It is not clear what this means. In the associated figure, Fig. 9, it appears that both the pulse rate and the pulse amplitude increases after cutting, which is counter to the statement in line 248. Please address this discrepancy.

Lines 249 and 250: Authors state: "No pulses were present after all leaves were cut even though they remained in the vicinity of the seagrass patch." This sentence must refer to a time well-later than shown in the data of Fig. 9. since neither the rate nor the amplitude of the pulses appear to be diminishing at the end of the plots in Fig. 9(B) or 9(E). How much longer were sounds produced, or in other words, how long did it take for the sounds to stop after cutting?

Figs 9(A), (B) and (C): The time axes on these spectrograms has only one tick mark labeled with time, hence it is difficult to relate the time axes of (A), (B) and (C) to the time axes of (D) and (E). Adding the markers where the cut was made is helpful but not sufficient. Please label more tick marks on the time axes of (A), (B) and (C).

Fig 9(C): What is the temporal relationship between the events in Fig. 9(B) and 9(C). There is no information about this in the text or in the caption. Was this after the end of the data in Fig. 9(B)? Please clarify.

Line 254: Authors state: "Kratochvil and Polliver (2017) reported gas bubble sounds from submerged American waterweed..." but in the paper Kratochvil and Polliver (2017), they report on Canadian waterweed. Please correct this in the present paper.

Line 256: Authors state: "...with most energy in the high-frequency range." This is too ambiguous. What is meant by high frequency range? Please state the frequency range explicitly.

Line 260: This comment is related to a previous comment about lines 235-236. The author have not proved that the emission of minute streams of bubbles is sound free. These minute bubbles are not the source of the lower-frequency sounds.


Comments about the Discussion section and the conclusions:

Authors invoke Roberts et al (1984), who conclude that the lacunae allows gas to flow through various parts of the plant, but not out of the plant unless various parts of the plant are cut. There are diaphragms in the lacuane but the diaphragms have holes. These holes allow the flow of air, but not of liquid. Roberts et al (1984) demonstrated this by pumping air through the plants an by directly observing where the gas emerged. I have no argument with this.

The present paper seems to suggest that airflow through these holes makes sound, and the present authors refer to the sound as a whistle citing Fletcher (1992) who describes whistling sounds due to six different arrangements associated with a turbulent jet. This reviewer is confused about this argument, because all of the sounds reported in the present paper are pulsed sounds, not whistle-like sound.

Another confusing thing is the present authors refer to air flow changing direction as it flows through the holes in the diaphragms as the cause for the sound and cite Fletcher (1992). Fletcher does not explicitly describe change of direction as the fundamental requirement for sound, rather Fletcher states that turbulence due to sufficiently high-velocity flow, when it interacts with various physical objects, creates whistle sounds. These arguments to not lead this reviewer to the same conclusions that the authors make:

"We demonstrate that photosynthesis increases leaf pressure driving oxygen through pores in diaphragms that separate portions of the plant." This reviewer does not believe that the present authors have demonstrated this. They did not measure pressure inside the plant, they observed sound that was dependent upon light.

The authors then state: "This gas movement produces a series of pulsed sounds that can vary in
pulse rate, amplitude and waveform." The authors do not directly observe any gas movement inside the plants as in Roberts et al (1984). The present authors do not provide any evidence of how internal flow that they claim behaves like Fletcher's whistles, but then somehow produces the pulsed sound they report. This is a fundamental problem. The authors have invoked a line of reasoning that explains entirely different sounds than the ones they report, hence this reviewer does not agree with their conclusions. A more general set of conclusions seems appropriate: Others have demonstrated that photosynthesis produces gas in plants and that can increase the pressure of the gas inside the lacunae of the plants. The present authors did not demonstrate that. Others have demonstrated that gas can flow through lacunae in the plants. The present authors did not demonstrate that. The present authors have observed pulsed sounds that are only produced when light is present and when photosynthesis has not been chemically suppressed. They have suggested that these sounds are associated with flow through the pores in the diaphragms in the lacunae, but they have not proved this. They report a remarkably constant pulse repetition rate, and various other metrics associated with a number of pulsed sound emissions, but have not successfully explained the origin of any of these sounds. The conclusions stated in the present manuscript must be rewritten accordingly.

Comments on the data file "peerj-101054-photosynthesis_pulsed_sound_recorded_at_dawn.wav":
This is a very interesting recording, but its origin is not explained in the manuscript or elsewhere, or at least this reviewer could not find that information, other than it was recorded at dawn (from the title of the file). I other words, I am not which experimental site or experimental preparation is associated with this recording. There were several described in the paper. This information should be provided. There are several segments where the pulse repetition rate and especially the amplitude of the pulses is remarkable constant. That is one aspect of this data which I think could be analyzed more completely and emphasized more. Others, including a paper cited in the present manuscript, Kratochvil & Pollirer (2017), report regular pulse amplitude, but the data here is even more regular.

Since this reviewer is not convinced of the fundamental conclusion of this paper, which is also stated in the title, and since this reviewer has found dozens of cases where significant ambiguities must be eliminated, a reject decision is recommended. A manuscript based on these measurements and analysis, with significant revisions to eliminate all the ambiguities, with a revised conclusion and a revised title is of interest to the community. I choose reject because the text associated with the "major revisions" decision is not strong enough. In the view of this reviewer, a "preference" for a re-evaluation is not sufficient. I believe review of a revised manuscript is mandatory.

---

## Round 0.2 · accepted · Accept

Dear Authors, thank you very much for the effort made in revising your manuscript. You have addressed all the reviewer's and my personal comments. Altough some minor corrections should be made according to the rev1 that re-read your answers, I feel confident that most of them should be made during the editing of the proofs.

Reviewer 1 ·

Basic reporting

Nice manuscript with some very interesting results. This definitely works better with fewer figures; the story is much clearer now. I also like the wider perspective of the results. However, I still think some further justification for the stats you did would be helpful for other researchers seeking to replicate your results.

I am also a bit confused about the decision to create three sub-categories of low pulse rates. I think there just needs to be a very clear rationale for doing this and why it’s ecologically / physiologically important.

Also, I was searching for a nice visual representation of three experimental treatments compared against each other. I think perhaps they are represented in Figure 4 but it’s not clear. Sorry if I’m missing something. Perhaps this could be made more obvious.

Experimental design

Good.

Validity of the findings

Good.

Annotated reviews are not available for download in order to protect the identity of reviewers who chose to remain anonymous.